# Zinc–Cobalt Oxide Thin Films: High Curie Temperature Studied by Electron Magnetic Resonance

**DOI:** 10.3390/molecules27238500

**Published:** 2022-12-02

**Authors:** Bogumił Cieniek, Ireneusz Stefaniuk, Ihor Virt, Roman V. Gamernyk, Iwona Rogalska

**Affiliations:** 1Institute of Materials Engineering, College of Natural Sciences, University of Rzeszow, Pigonia 1, 35-310 Rzeszow, Poland; 2Institute of Physics, College of Natural Sciences, University of Rzeszow, Pigonia 1, 35-310 Rzeszow, Poland; 3Institute of Physics, Mathematics, Economy and Innovation Technologies, Drohobych Ivan Franko State Pedagogical University, Stryiska 3, 82100 Drohobych, Ukraine; 4Department of Experimental Physics, Ivan Franko National University of Lviv, 79005 Lviv, Ukraine

**Keywords:** ferromagnetism at room temperature, electron magnetic resonance, low-field microwave absorption, zinc oxide, silver nanoparticles

## Abstract

The material with a high Curie temperature of cobalt-doped zinc oxide embedded with silver-nanoparticle thin films was studied by electron magnetic resonance. The nanoparticles were synthesized by the homogeneous nucleation technique. Thin films were produced with the pulsed laser deposition method. The main aim of this work was to investigate the effect of Ag nanoparticles on the magnetic properties of the films. Simultaneously, the coexisting Ag^0^ and Ag^2+^ centers in zinc oxide structures are shown. A discussion of the signal seen in the low field was conducted. To analyze the temperature dependence of the line parameters, the theory described by Becker was used. The implementation of silver nanoparticles causes a significant shift of the line, and the ferromagnetic properties occur in a wide temperature range with an estimated Curie temperature above 500 K.

## 1. Introduction

Zinc oxide (ZnO) is an inexpensive well-known semiconductor with potential in various applications [1,2,3], such as varistors [4] or sensors [5,6]. It is important that some of the zinc can be replaced by magnetic transition metal ions (TM) to create a metastable solid solution. Since both Zn^2+^ and Co^2+^ ions have nearly identical ion radii, doping ZnO with cobalt is most interesting [7]. Furthermore, parameters such as piezoelectricity and transparency in the visual region have attracted great interest from researchers in ZnO-based diluted magnetic semiconductors (DMS) [8,9] due to their possible technological applications in spintronics [10,11,12,13]. Using the modified Zener model, Dietl et al. suggested that *p*-type DMS based on ZnO could lead to a transition temperature greater than room temperature [14]. In this theory, *p*–*d* interactions are the cause of long-range magnetic coupling, but the studied ZnO samples are either insulating or conducting *n*-types. Some theoretical works using density functional theory (DFT) [15,16] show that *n*-type cobalt-doped ZnO shows ferromagnetism (FM) at room temperature (RTFM). Some research groups reported ferromagnetism in ZnO doped with transition metals with Curie temperatures (*T_C_*) from 30 to 550 K [17,18,19,20,21,22,23], and some found antiferromagnetic, spin-glass, or paramagnetic behavior [9,24,25]. The existence of a ferromagnetic order in Co-doped ZnO is suggested to be attributed to double exchange [8] or the Ruderman–Kittel–Kasuya–Yosida (RKKY) interaction between Co ions [26]. Theoretical calculations show that ground-state ZnO with Co ions is spin-glass because of the short-range interactions between TM atoms [27].

ZnO doped with gold (Au) or silver (Ag) increases the photocatalytic activity of the composite by reducing electron-hole recombination and improving separation [28]. Silver nanoparticles (NP) have been investigated by many scientists because of their significant role in applications of visible light absorption [29,30]. Many works have described the synthesis nanocomposites of heterogeneous ZnO/Ag via a variety of synthetic routes for various applications, such as disinfection and wastewater treatment [31,32,33,34,35,36,37,38,39]. In addition, research confirms that the existence of Ag NPs on the ZnO surface reduces the intensity of electron magnetic resonance (EMR) signals and may lead to improved photodegradation efficiency [40].

Obtaining a homogeneous thin layer with *p*-type ZnO is a challenging task. One of the proposed methods is the addition of silver ions [41,42,43]. The pulsed laser deposition (PLD) method has a wide range of particle energies, allowing mainly Zn ions to penetrate deeper into the substrate, forming a mixed structure with the desired conductive type [43]. The main aim of this work was to investigate the influence of silver nanoparticles on ZnO doped with cobalt-ion thin film and to determine the changes in magnetic properties compared to that of layers without silver NPs.

## 2. Results and Discussion

Measurements of EMR were taken on samples with a quartz and silicon substrate. Angular dependence measurements of the Zn_0.8_Co_0.2_O/Ag film on the quartz substrate were measured at room temperature; a summary is shown in Figure 1.

The resulting angular dependence is characteristic of magnetic layers and the combination of magnetic and nonmagnetic layers. A strong anisotropy of the spectrum is observed, along with a change in the shape of the line, with the result that, at certain angles, the width of the line increases significantly as the intensity decreases, and the EMR line is no longer visible. However, a line visible all the time in the low field remains, the so-called low-field microwave absorption (LFMA), which is also called an indicator of FM properties for a large group of materials. The LFMA signal for ferrites and magnets is related to the beginning of the ordered phase and is a sensitive detector of magnetic ordering [44,45]. For soft magnetic materials, the signal is due to low-field processes of spin magnetization [46].

EMR measurements as a function of temperature were performed in two temperature ranges, from 300 to 500 K and from 97 to 300 K. Figure 2 shows the EMR spectra of Zn_0.8_Co_0.2_O/Ag as a function of temperature in the range from 300 to 500 K. The measurement was performed at an angle when the EMR line was close to its extreme position, and it corresponds to an orientation of 115 degrees from the angular dependence in Figure 1.

In contrast to the angular dependence of the EMR spectrum, the temperature dependence over the entire temperature range studied does not show large changes in the shape and position of the EMR line. We can see a broad line moving in the direction of the low magnetic field for Zn_0.8_Co_0.2_O/Ag compared to the sample without silver NPs (Figure 3) and to layers of Co-doped ZnO (in our previous papers [47,48,49]).

Such a large shift of the EMR line can be explained by the appearance of ferromagnetic properties in Zn_0.8_Co_0.2_O embedded with Ag NPs. A similar effect is observed in many works; for example, in spin-glasses or soft and hard magnetic layers [50,51,52,53]. The hard and soft layers of the spring magnets are coupled at the interfaces due to the strong exchange coupling between them. A high magnetic saturation is achieved by a soft magnet, whereas a high coercivity field is achieved by the magnetically hard material. The shift of the EMR line towards a low magnetic field for the layers of work [52] is related to the layer thickness and occurs when the thickness changes from 10 to 20 nm. In our sample, we observe conglomerates of nanoparticles with sizes in the order of 80 nm, as well as single nanoparticles, so, in addition to the EMR line, we also observe a spectrum from single silver nanoparticles. Low-intensity lines can also be seen in a field of about 340 mT. These were assigned to silver ions Ag^0^ (4d^10^5s^1^) and/or Ag^2+^ and Ag^0^ (4d^9^) (Figure 4) (described in the literature [54]). This confirms that Ag^0^ and Ag^2+^ centers can coexist simultaneously in zinc oxide structures, with Ag^+^ being inactive in the EMR signal. Matching the EMR spectrum with the Dyson-type line results from a good fit to our sample Zn_0.8_Co_0.2_O (Figure 4), and hence the line parameters were obtained: the peak-to-peak linewidth (*H_pp_*), the EMR intensity (*I*), and the resonance field (*H_r_*). Xepr software was used to analyze and determine the parameters of the EMR line; this is the standard EPR spectrometer software used to control and analyze the spectrum.

Figure 5 shows the intensities of the EMR line as a function of temperature in two temperature ranges.

The nature of the temperature dependence of the line parameters (*H_pp_*, *I*, and *H_r_*) suggests that the *T_C_* is higher than 500 K. The theory described by Becker [55] was used to analyze the temperature dependence (where *T* is the temperature of the measurement). Becker calculated the EMR resonance field shift and linewidth as a function of temperature and frequency near freezing temperature for spin-glass alloys, using RKKY exchange coupling and a smaller anisotropic interaction. To fit our line parameters’ behavior, we adopted Becker’s theory for the critical regime (*T~T_C_*) and spin-glass regime (*T < T_C_*). In Zn_0.8_Co_0.2_O/Ag, we can see an abnormal reduction in the linewidth resonance (*H_pp_*) with a minimum at or near the critical temperature (Figure 6).

To fit the linewidth about the minimum, we used the function [50,51,55]:(1)$$\Delta H=a_{0}+b\prime \;{\left|\frac{T-T_{min}}{T_{min}}\right|}^{n}\;$$
where *a*_0_ is the residual linewidth; *b*′ is the thermal broadening constant (independent of the orientation of the static field); *n* is the exponent of the expression for the length of the correlation associated with the distributed magnetization of Huber’s theory of linewidth in isolated ferromagnets and antiferromagnets in the region near the critical temperature [56,57]; and *T_min_* represents the temperature of the minimum linewidth. The large value of the exponent *n* = 3 *v*/2 that we obtain is consistent with the presence of a strong perturbation, which is expected to increase the rate at which the correlation length decreases as the temperature moves away from the critical temperature. In the theory of the mean-field 3D Heisenberg model *v* = 0.71, and the value that we obtained was *v* = 1.33 [58]. The best fit of Equation (1) is shown in Figure 6 as the black line, where *a*_0_ = 97.7 mT, *b′* = 89 mT, *T_min_* = 380 K, and *n* = 2.0. This agrees with experimental data in the temperature range of 300 to 500 K, and the value of *a*_0_ = 97.7 mT implies that the effects of the crystal field and demagnetization in Zn_0.8_Co_0.2_O/Ag are high. The origin of the residual linewidth component in the spin-glass alloys has been assigned to local moment imperfections and crystal-field effects via a mechanism of demagnetization.

In the low-temperature regime (*T* < *T_C_*), we can see a linewidth broadening and a shift in the resonance field. This is similar to the dependence in spin-glass alloys (for example, AgMn_x_Sb_y_) and in molecule-based magnets [50,55]. In these alloys, the excess linewidth was assigned to an exchange-narrowed anisotropic interaction. With the decreasing temperature, the slowing down of the spin fluctuations reduces the effectiveness of the exchange-narrowing. Moreover, these alloys also show a related shift in the resonance field, which is neither a frequency-independent internal field nor a pure g-shift. Becker calculated the EMR linewidth and line shift effects for spin-glasses with anisotropy [55]. For systems with no remnant magnetization, Becker has shown that the resonance field and linewidth are given by formulas [50,51,55]:(2)$$\Delta H=\frac{ABT}{B^{2}+T^{2}}$$
and
(3)$$H_{r}=H_{0}+\frac{AT^{2}}{B^{2}+T^{2}}$$
where $A=\frac{g\mu_{B}K}{\hbar \omega \chi_{⊥}}$, $B=\frac{M_{2}}{Kk_{b}\omega }$, $H_{0}=\frac{\hbar \omega }{g\mu_{B}}$; and ω is the resonance frequency. Here *K* is the constant of anisotropy, $\chi_{⊥}$ is the static susceptibility of transverse, and *M*_2_ is associated with spin relaxation.

The best fits to the experimental data (shown in Figure 7 and Figure 8) are *A* = 39.62 mT, *B* = 44.49 K, and *H*_0_ = 19.12 mT. Very good agreement between the experimental data and the fitting (Equations (2) and (3)) is found. The linewidth and resonance field shift are compatible with a disordered ferrimagnet near and below critical temperature. Additional confirmation of the observed ferromagnetic properties is provided by the LFMA line. The absorption of microwave power centered at zero magnetic field has been reported in ferromagnetic materials and in various other materials, such as ferrites, high-temperature superconductors, and soft magnetic materials [44,46,59]. For soft magnetic materials, the LFMA signal is induced by low-field processes of spin magnetization [46]. The appearance of LMFA lines is an indicator of the ferromagnetic properties of the material. Figure 9 shows the magnetic hysteresis that is very often peeled off in the literature for LFMA lines.

A similar hysteresis of the LFMA line was observed for Zn_0.8_Co_0.2_O/Ag on a silicon substrate, shown in Figure 10.

For the Zn_0.8_Co_0.2_O/Ag film on a silicon substrate, we observe a lower intensity of the EMR lines and LFMA lines, but the nature of the observed magnetic properties is similar. The addition of gold produces a similar effect, although with a weaker intensity. The same layer deposited on a silicon substrate produces an analogous effect, including a hysteresis loop, although with a weaker intensity. The observed changes in the EMR spectrum obtained in both directions of registration show hysteresis only near the zero magnetic field; meanwhile in the rest of the range, the spectrum has an identical form and we do not observe changes in the shape of the EMR spectrum as presented in the paper [60].

## 3. Materials and Methods

Electron magnetic resonance measurements in the continuous wave X-band were taken on the Bruker FT-EPR ELEXSYS E580 spectrometer (Bruker Analytische Messtechnik, Rheinstetten, Germany). To control the temperature, Bruker liquid nitrogen cryostats were used with the 41131 VT digital controller, and the angular dependences of the EPR spectra were performed using a one-degree programmable goniometer E218-1001 (Bruker Analytische Messtechnik, Rheinstetten, Germany).

Samples of ZnO doped with cobalt thin films (Zn_1−*x*_Co*_x_*O, *x* = 0.2) embedded with Ag NPs were obtained using a combination of PLD and homogeneous nucleation techniques. A homogeneous nucleation technique was chosen for the synthesis of silver nanoparticles (more details on the creation of noble metal NPs are given in the work [61]). A droplet of aqueous Ag NPs was then deposited on the surface of the substrate with further drying under surrounding conditions. For the PLD method, a silicon and quartz substrate was chosen. The KGd(WO_4_)_2_ laser was used—radiation characteristics: *λ* = 1067 nm, beam energy density 6–8 J/cm^2^, repetition rate 10–0.3 Hz, pulse duration *t* = 20 ns. The technology module used the Q-switch to irradiate in modulated goodness factor mode. The deposition temperature (substrate temperature) was about 200 °C. The thickness of the resulting layer was about 300 nm. Therefore, a planar nanocomposite consisting of a Zn_0.8_Co_0.2_O thin film deposited on silver nanoparticles (Zn_0.8_Co_0.2_O/Ag) was formed. The specificity of the PLD method is the wide particle energies spread, so we expect a deeper penetration of the applied particles into the substrate and the formation of a complex structure at the contact zone of the resulting layer and Ag NPs.

The basic parameters of the obtained layers were controlled, and the results are included in the Appendix A.

## 4. Conclusions

We have performed X-band EMR studies of the Zn_0.8_Co_0.2_O embedded with Ag NPs. A line in a broad asymmetric Dyson shape associated with magnetic interactions was observed. The results of a temperature dependence analysis for the EMR linewidth and resonant field based on Becker’s model were developed. From the critical regime (*T*~*T_C_*), the temperature *T_min_* = 380 K was determined for the minimum width of the EMR linewidth. The value of *a*_0_ = 97.7 mT suggests that demagnetization and the effects of the crystal field in Zn_0.8_Co_0.2_O/Ag are high. For the range of the low-temperature regime (*T* < *T_C_*), constants *A* and *B* related to the magnetic properties were determined. *A* = 39.62 mT, *B* = 44.49 K, and *H*_0_ = 19.12 mT. Very good agreement between the experimental data and the fitting was found. The linewidth and resonance field shift are in accordance with a disordered ferrimagnet near and below the critical temperature. In addition, small, intense lines are assigned to the silver ions Ag^2+^ and Ag^0^ [54]. There is a shift of the line towards the low-field direction for samples with silver nanoparticles.

The aim of this study was to investigate the effect of silver nanoparticles on the magnetic properties of Zn_0.8_Co_0.2_O. It was shown that the implementation of silver nanoparticles causes a significant shift of the ferromagnetic resonance (FMR) line (Figure 3) and that the ferromagnetic properties occur in a wide temperature range with an estimated Curie temperature above 500 K; the description is consistent with the adopted model, which can be considered as a major achievement of this work. Based on the results obtained, it can be claimed that a material with desirable magnetic properties at room temperature has been obtained, with potential applications that are important in spintronics.

## Figures and Tables

**Figure 1 molecules-27-08500-f001:**
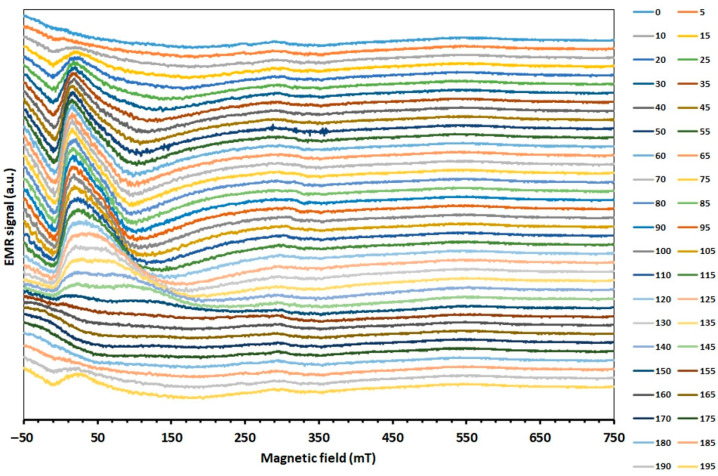
EMR spectra as a function of the angular dependence of Zn_0.8_Co_0.2_O/Ag film.

**Figure 2 molecules-27-08500-f002:**
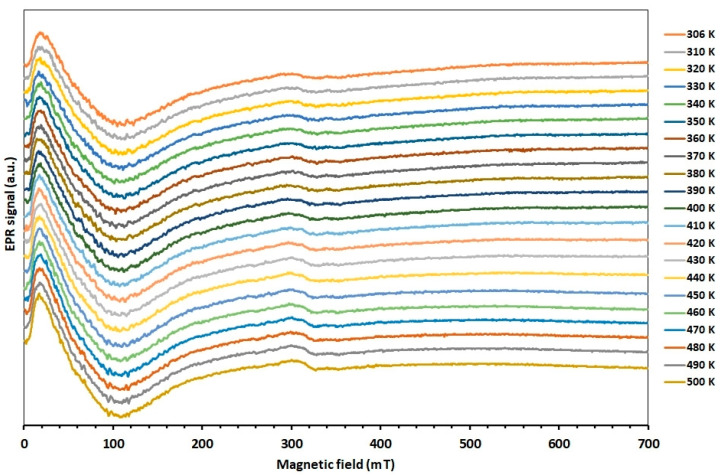
EMR spectra of Zn_0.8_Co_0.2_O/Ag as a function of temperature in the range from 300 to 500 K.

**Figure 3 molecules-27-08500-f003:**
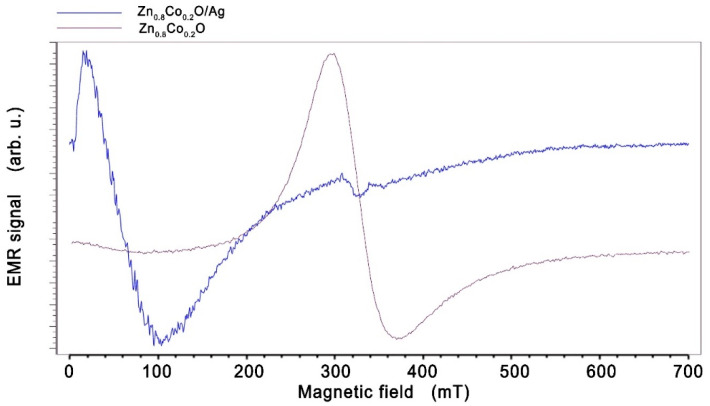
EMR spectra of Zn_0.8_Co_0.2_O (purple) and Zn_0.8_Co_0.2_O embedded with Ag NPs (blue), obtained at temperature 160 K.

**Figure 4 molecules-27-08500-f004:**
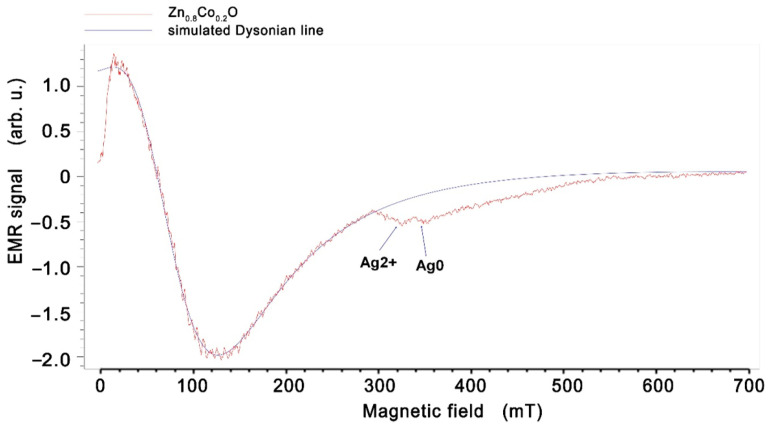
EMR spectra of Zn_0.8_Co_0.2_O embedded with Ag NPs film obtained at temperature 300 K with the simulated Dyson-type line. Small intensity lines from Ag^2+^ and Ag^0^ are described.

**Figure 5 molecules-27-08500-f005:**
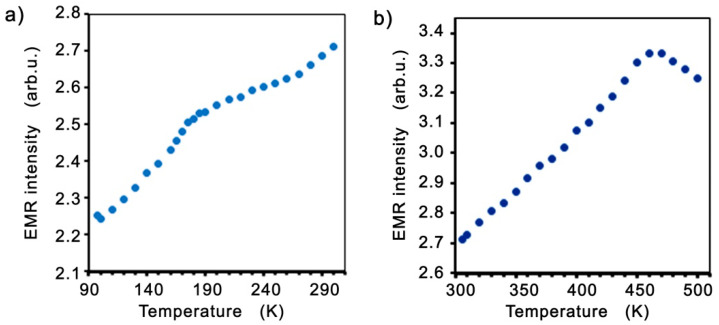
The EMR line intensities as a function of temperature in two temperature ranges, below (**a**), and close to *T_C_* (**b**).

**Figure 6 molecules-27-08500-f006:**
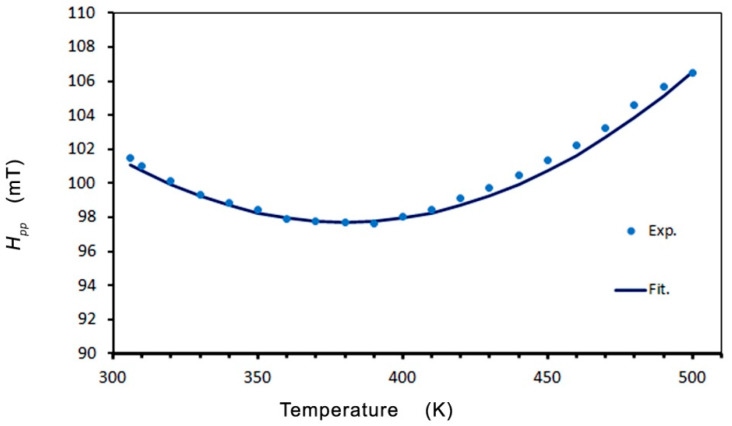
Temperature dependence of the resonance linewidth *H_pp_*.

**Figure 7 molecules-27-08500-f007:**
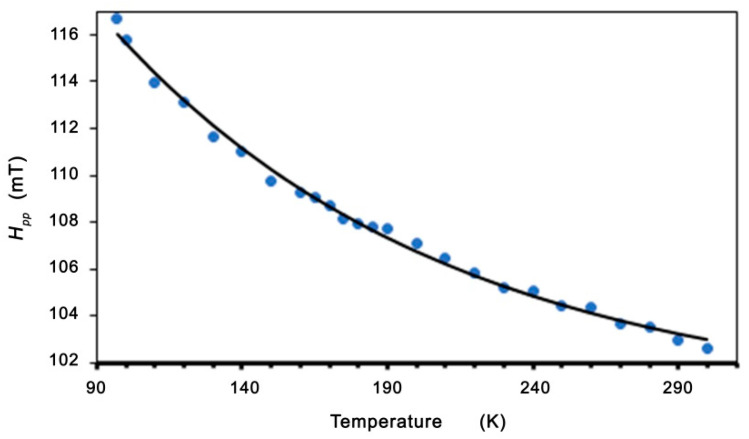
Temperature dependence of the peak-to-peak linewidth for Zn_0.8_Co_0.2_O/Ag. Theoretical line fitted based on Equation (2).

**Figure 8 molecules-27-08500-f008:**
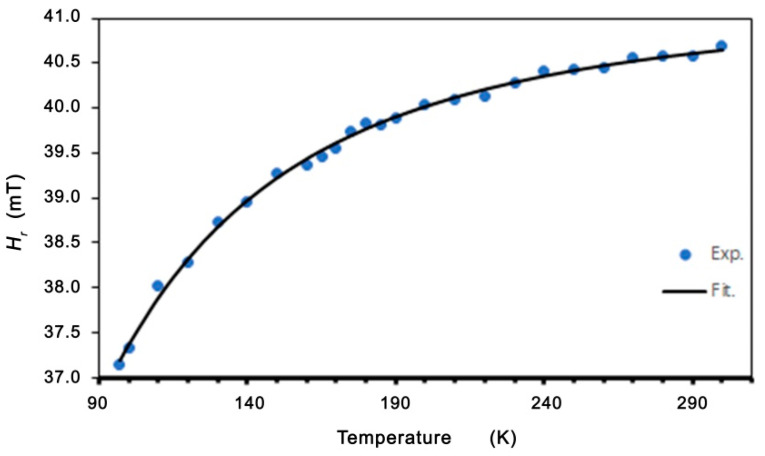
Temperature dependence of the resonance field for Zn_0.8_Co_0.2_O/Ag. Theoretical line fitted based on Equation (3).

**Figure 9 molecules-27-08500-f009:**
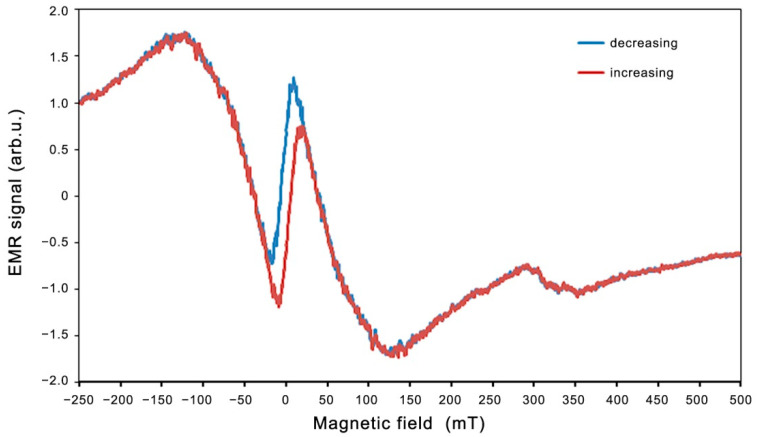
EMR spectrum recorded for an increasing (red) and decreasing (blue) magnetic field at 300 K for Zn_0.8_Co_0.2_O/Ag on a quartz substrate.

**Figure 10 molecules-27-08500-f010:**
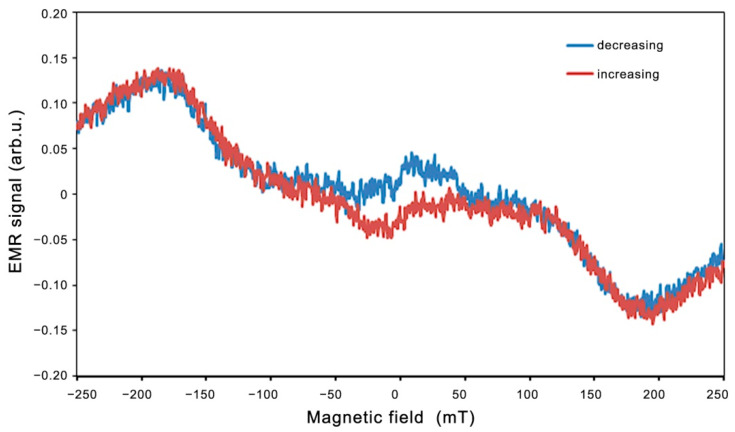
EMR spectrum recorded for an increasing (red) and a decreasing (blue) magnetic field at 140 K for Zn_0.8_Co_0.2_O/Ag on a silicon substrate.

## Data Availability

Data available in a publicly accessible repository.

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
