# Peer review of "Zinc–Cobalt Oxide Thin Films: High Curie Temperature Studied by Electron Magnetic Resonance"

_molecules, 2022, doi:10.3390/molecules27238500_

Round 1

Reviewer 1 Report

The manuscript reports cobalt-doped zin oxide embedded with silver nanoparticles thin films with the studies of the effect of Ag nanoparticles on the magnetic properties of films via electron magnetic resonance measurements. The research is systematically done on a good level, the conclusions are convincing. I recommend therefore its publication in Molecules with major revisions.

1.       The style of figures should be consistent. The figures are poorly drawn and need to be redraw refering to the style of figure in the paper published in Molecule.

2.       In page 3, line 101-103, The angular of EMR measurements as a function of temperature need to be given.

3.       In Equation (1), the meaning of ‘b′’ and ‘n’ need to be explained.

4.       The reason to use Equation (1) needs to be given. Any reference?  What physics can be known from the fitting?

Author Response

Thank you very much for the review of our work entitled: "Zinc Cobalt Oxide Thin Films: High Curie Temperature Studied by Electron Magnetic Resonance". Thanks to the review, we are able to improve the quality of our work. Below are our responses to the point brought up by the reviewer.

  1. The style of figures should be consistent. The figures are poorly drawn and need to be redraw refering to the style of figure in the paper published in Molecule.

The style of drawings has been unified and improved. The resolution of all the figures has been improved. On Fig. 3 the y-axis has been removed (because the scale is different to other figures), and also signatures for Figs. 9 and 10 were improved.

  1. In page 3, line 101-103, The angular of EMR measurements as a function of temperature need to be given.

In line 103, a new sentence was added: “The measurement was performed at an angle when the EMR line is close to its extreme position and it corresponds to an orientation of 115 degrees from the angular dependence in Figure 1.”

  1. In Equation (1), the meaning of ‘b′’ and ‘n’ need to be explained.
  2. The reason to use Equation (1) needs to be given. Any reference?  What physics can be known from the fitting?

Answer for both problems: New references have been added. We also added references to the equations Eq.1 and Eq. 2 ([51,52,56]).

In line 153, we expand the sentence to include new information: ‘...where a0 is the residual linewidth, b’ is the thermal broadening constant (independent of the orientation of the static field), n is the exponent of the expression for the length of the correlation associated with the distributed magnetization from Huber's theory of line width in isolated ferromagnets and antiferromagnets in the region near the critical temperature [57,58], and Tmin represents the temperature of the minimum linewidth. The large value of the exponent n=3ν/2 that we obtain is consistent with the presence of a strong perturbation, which is expected to increase the rate at which the correlation length decreases as the temperature moves away from the critical temperature. In the theory of the mean field 3D Heisenberg model v = 0.71 and the value we obtain ν =1.33 [59].”

Reviewer 2 Report

The paper “Zinc Cobalt Oxide Thin Films: High Curie Temperature Studied by Electron Magnetic Resonance” by BogumiÅ‚ Cieniek et al. devoted to the study of influence of silver nanoparticles (Ag NPs) on the properties of the cobalt-doped zinc oxide (ZnO) thin films.  be electron magnetic resonance (EMR) by using electron paramagnetic resonance (EPR) spectrometer. It is shown that both Ag0 and Ag2+ containing NPs are embedded into the film. It is shown that silver incorporation leads to the significant shift of EMR line. Curie temperature is derived from the EMR measurements.

Doped ZnO is from one side is a very well-studied material, but from other side it has many undiscovered or not well-defined (discussable) properties like p- or n- type of conductivity being semiconductor; ferromagnetic or antiferromagnetic, etc. Playing with dopants and technologies of synthesis, one can get various species with a variety of properties. In this work pulsed laser deposition (PLD) and homogeneous nucleation techniques were used to obtain p-type ZnO with cobalt thin films and Ag NPs.

A lot of experimental work is done. The paper sounds and is deserved to be published in Materials.

Comments and remarks.

1.     Conclusions. Line 214 (as well as line 181). I did not get the meaning of the sentence “Very good agreement was found”. Agreement between which parameters or which opinion? Between the experimental data and fitting?

2.      Figure 6. Is there any dependence of the linewidth on the direction of the external magnetic field sweep (normal way – from the low fields to the high field and unusual way – from the high field to the low value, as in Figures 9 and 10) or modulation depth? In a series of works by Ravkin et al., Talanov et al., Kurkin et al. it was shown that modulation and direction of the sweep does matter and, therefore, the model of the “frozen” spin-fluctuations for the high-temperature superconductors, for example, should be revised. There it was explained by the pinning of vortexes.
[https://iopscience.iop.org/article/10.1088/0953-2048/18/9/007 ]

3.     Can the homogeneity of the sample be measured by EMR? Whether it is possible to give some numeric parameter for that from EMR?

Author Response

Thank you very much for the review of our work entitled: "Zinc Cobalt Oxide Thin Films: High Curie Temperature Studied by Electron Magnetic Resonance". Thanks to the review, we are able to improve the quality of our work. The style of drawings has been unified and improved. The resolution of all the figures has been improved. On Fig. 3 the y-axis has been removed (because the scale is different to other figures), and also signatures for Figs. 9 and 10 were improved.

Below are our responses to the point brought up by the reviewer.

  1. Line 214 (as well as line 181). I did not get the meaning of the sentence “Very good agreement was found”. Agreement between which parameters or which opinion? Between the experimental data and fitting?

In lines 181 and 214, a new sentence was added: “Very good agreement between the experimental data and the fitting (Eq. 2 and Eq. 3) is found.”

  1. Figure 6. Is there any dependence of the linewidth on the direction of the external magnetic field sweep (normal way – from the low fields to the high field and unusual way – from the high field to the low value, as in Figures 9 and 10) or modulation depth? In a series of works by Ravkin et al., Talanov et al., Kurkin et al. it was shown that modulation and direction of the sweep does matter and, therefore, the model of the “frozen” spin-fluctuations for the high-temperature superconductors, for example, should be revised. There it was explained by the pinning of vortexes. [https://iopscience.iop.org/article/10.1088/0953-2048/18/9/007 ]

We thank you for pointing out an interesting article, but in our case we do not observe the described effects. This is especially evident in Fig. 9, where only near zero magnetic field we observe a hysteresis of the spectrum, after that the EMR line waveforms in both directions coincide.

We added a sentence to the paper in the line 203: “The observed changes in the EMR spectrum obtained in both directions of registration show hysteresis only near zero magnetic field, while in the rest of the range the spectrum has an identical form and we do not observe changes in the shape of the EMR spectrum as presented in the paper [61]."

  1. Can the homogeneity of the sample be measured by EMR? Whether it is possible to give some numeric parameter for that from EMR?

In principle, homogeneity cannot be measured in classical terms with typical EMR measurements, but for crystalline (solid) samples, the anisotropy of the paramagnetic center and interactions, can be studied.

Reviewer 3 Report

L147  Possessive:  Becker's not Beckers.

Fig. 9,10.   Why are the words "left" and "right" needed in identifying the spectra?  Why not refer to them by their colors? 

Figs.   Are all the figures with EMR Intensity (a.u.) presented with the same a.u. or are they scaled differently?

More details are needed in the Materials & Methods section.  The unfamiliar reader might wish to know why an EPR spectrometer is used to perform EMR.  I believe the Bruker ELEXSYS E580 has pulsed and CW capabilities.  What mode was used for this study?  I was not aware that the Bruker EPR could scan up and down in field.  Is this a common feature or a modification?  What cavity was used with the cryostat?

Author Response

Thank you very much for the review of our work entitled: "Zinc Cobalt Oxide Thin Films: High Curie Temperature Studied by Electron Magnetic Resonance". Thanks to the review, we are able to improve the quality of our work. The style of drawings has been unified and improved. The resolution of all the figures has been improved. On Fig. 3 the y-axis has been removed (because the scale is different to other figures), and also signatures for Figs. 9 and 10 were improved.

Below are our responses to the point brought up by the reviewer.

L147  Possessive:  Becker's not Beckers.

Thank you, it has been corrected in paper.

Fig. 9,10.   Why are the words "left" and "right" needed in identifying the spectra?  Why not refer to them by their colors? 

We added in the description the direction of sweeping with color names.

“Figure 9. EMR spectrum recorded for an increasing (red) and decreasing (blue) magnetic field at 300 K for Zn0.8Co0.2O/Ag on a quartz substrate.”

“Figure 10. EMR spectrum recorded for an increasing (red) and a decreasing (blue) mag-netic field at 140 K for Zn0.8Co0.2O/Ag on a silicon substrate.”

Figs.   Are all the figures with EMR Intensity (a.u.) presented with the same a.u. or are they scaled differently?

Yes, all EMR spectra (arb.u.) are in the same scale. The only exception is Fig. 3, from which we have removed the y-axis values.

More details are needed in the Materials & Methods section.  The unfamiliar reader might wish to know why an EPR spectrometer is used to perform EMR. 

From a technical point of view, EPR and EMR are the same technique. The differences are due to the variety of materials and interactions studied, so since our work deals with magnetic materials, it is most appropriate to use the abbreviation EMR.

I believe the Bruker ELEXSYS E580 has pulsed and CW capabilities.  What mode was used for this study?  I was not aware that the Bruker EPR could scan up and down in field.  Is this a common feature or a modification?  What cavity was used with the cryostat?

Yes, our Bruker ELEXSYS E580 has both pulsed and CW capabilities, with X- and Q-band. The direction of the field sweep, up and down, is a feature in our equipment. For X-band we are using the ER 4123D cavity. The main measurement parameters are: modulation frequency 100 kHz, and modulation amplitude 1G.

In line 62: we added the “continuous wave”. Information about the X-band was already there.

Round 2

Reviewer 1 Report

The manusript can be accepted now.